# Association of lifestyle and behavioral factors with self-reported visual problems among schoolchildren in rural Bangladesh

Fakir M Amirul Islam[1,2]*, Arzan Hosen[2], Abdullah Al Mahmud[3]

**1** School of Health Sciences, Swinburne University of Technology, Australia, **2** Organization for Rural Community Development (ORCD), Narail, Bangladesh, **3** Centre for Design Innovation, School of Design and Architecture, Swinburne University of Technology, Australia

* fislam@swin.edu.au

## Abstract

### Background

Lifestyle, environmental, and genetic factors influence visual impairment. The current study aims to report the sociodemographic, lifestyle, and behavioral factors associated with self-reported ocular conditions among children.

### Materials and methods

Data were collected from 13341 children aged 7–14 from 176 primary schools and 16 madrasas (Islamic educational institutes) of Narail Upazila from 30 November 2022 to 20 August 2023. Data on sociodemographic factors, including living conditions and parents' education; lifestyle and behavioral characteristics, including vitamin A consumption, watching TV or mobile; and self-reported ocular conditions, including seeing the blackboard or distant people, were collected. Chi-square tests and logistic regression analyses reported the association between sociodemographic characteristics with lifestyle and behavioral characteristics, and self-reported ocular conditions. The statistical software SPSS was used for data analysis.

### Results

Of the total children, 52.5% were girls. Almost 99% had taken Vitamin A, 59% watched TV or mobile screens regularly, 99 (0.7%) children reported that they had problems seeing the blackboard or distant people, only 59 (0.4%) children had eye examinations previously, and 32 (0.2%) children used spectacles even if they had experienced any adverse ocular conditions. The proportion of children watching TV (67% vs. 58%) or mobile screens (69.6% vs. 57.6%) was higher in urban than rural areas. The proportion of children who had problems seeing blackboard was higher in urban areas (2.2% vs. 0.6) than in rural areas and among mothers

**Data availability statement:** All data are in the manuscript and/or Supporting Information files.

**Funding:** I received a fund from Excellence in Vision Award (XOVA).

**Competing interests:** The authors have declared that no competing interests exist.

with higher education (3.1% among graduate mothers vs. 0.3% among mothers without schooling). More than 99% of children had no eye examination before this screening program. Watching TV one-hour relative risk 4.16, (95% confidence interval (CI): 2.15–8.07 or more than one hour RR 5.33, 95% CI: 2.62–10.8 was associated with a higher proportion of seeing problems than those who did not watch TV. Those who used mobile for one hour, 4.82, (95% confidence interval (CI): 2.17–10.7 or more than one hour RR 7.30, 95% CI: 3.06–17.4 was associated with a higher proportion of reporting any ocular trauma than who did not use mobile phone.

## Conclusion

Vitamin A taken among schoolchildren is very high, and self-reported ocular problems are minimal. Children living in urban areas are more prone to behavioral risk factors for visual impairment and have a higher proportion of ocular trauma. Increased awareness of vision impairment and its risk factors by schoolteachers can be a feasible and cost-effective approach to improving eye health in schoolchildren, especially in resource-poor setting.

## Introduction

Visual impairment in children refers to all degrees of reduction in vision in childhood, which can significantly affect the development of visual, motor, and cognitive function and potentially lead to long-term adverse psychosocial consequences [1–3]. According to WHO estimates, at the beginning of the VISION 2020 program, about 19 million children under the age of 15 years were visually impaired, and 1.4 million children had irreversible blindness. It was predicted that half of the blindness cases were preventable [4]. The prevalence of blindness in low and middle-income countries, including Bangladesh, ranges from 0.2 to 7.8/10,000 people, and in developed and industrialized countries, the annual incidence is 6/10,000 in the under-15 age group [5]. In school, visual impairment has been shown to adversely affect a child's educational performance, including reading skills [6–8]. Approximately 90% of visually impaired children do not receive adequate education due to factors including discrimination, stigmatization, and lack of access to appropriate schools [9]. Reports suggest that most VI is preventable or treatable [10–12]. However, there is data that only 4% of rural people have had an eye check once in their lifetime [13]. Understanding the lifestyle and behavioural factors associated with ocular problems, early detection, and effective methods of addressing the factors are essential approaches for preventing visual impairment [14, 15].

The risk factors of visual impairment include watching TV, mobile phone, family history of ocular conditions, and Vitamin Deficiency [16–19]. Vitamin A, or all-trans-retinol, is an essential micronutrient for immunity, the maintenance of mucosal surfaces, and vision; thus, its deficiency is associated with visual

impairment [19]. Periodic, high-dose vitamin A supplementation is a proven, low-cost intervention and the estimated number of saving lives of more than 350,000 children each year [20]. However, problematic smartphone use (PSU) has been becoming a challenging health issue for children as it has severe adverse effects on their mental health, emotional stability, self-control, and vision [21, 22]. The number of smartphone users is consistently increasing, reported to be 1 billion in 2014 and 5.5 billion in 2024 and projected to be 6.5 billion in 2029 globally, with a 14.9% annual growth rate [23].

Bangladesh is a low-medium income country in southeast Asia with an estimated number of people of 169.8 million, including 56.9 million children aged 0–17 years [24]. In Bangladesh, mobile phone use is increasing at one of the highest rates in the world. According to the Bangladesh Telecommunication Regulatory Commission (BTRC) report in 2020, around 74% of Bangladesh's 15–65-year-old population had mobile phones, with gaps between ownership and accessibility, including urban and rural areas, socioeconomic status, and occupation [25]. An overall increase in smartphone use was 8.59% from 2021 to 2024, but there was a decrease in TV ownership [26]. Due to the increase in access to mobile phones and internet, children are expected to have more access to mobile facilities. To develop a successful visual impairment prevention program, it is first necessary to characterize the socio-demographic, lifestyle, and behavioral factors and their association with ocular problems [3]. Besides the vitamin A supplement, it is now prime time to undertake strategic policies considering the findings for limiting children's smartphone usage, making Bangladesh susceptible to protecting its future generation from the harmful effects of mobile phones.

Some programs exist at the school level, including health programs funded by the Bangladesh government that perform periodic vision screening and Vitamin A supplements for each child [27]. Vitamin A supplement has been a successful program. In 2022, vitamin A supplement was reported to be 92%, a positive indication of preventing vision impairment. Children are primarily unaware and cannot always point out vision deficiency at an early stage, and parents may remain unaware of the gradually developing vision problem. Islam et al. previously reported that 96% of rural people did not have any eye examination in their lifetime and were unaware of their vision condition. Similar to the successful A supplement program, no such outcome related to vision screening has been reported, and the impact of the government/s program on vision screening is not well understood. Alam et al. studied the ocular status, health-seeking behaviors, and barriers to the uptake of eye care services among children of a slum community [28]. The study among 410 children aged 5–16 reported that 36.6% had ocular abnormalities, including 26.7% refractive error, of whom 73% never went to an eye care specialist. Epidemiological data on the risk factors of vision impairment, current symptoms of vision problems, and their associations with self-reported ocular problems in children would be beneficial for informing policy and planning control strategies. The current study aimed to report (i) the association of sociodemographic factors with lifestyle and behavioral factors and self-reported ocular problems and (ii) the association of lifestyle and behavioral factors with self-reported ocular problems among children.

## Materials and methods

### Study design

We conducted this cross-sectional study between 30 November 2022 and 20 August 2023 in the Narail district in Bangladesh. To understand the administrative subdivision, Bangladesh, a country of 160 million people, is divided into 64 districts. Each district is divided into sub-districts named Upazilas (493 in total), and each Upazila is further divided into several Unions and the city center known as pourashava. Each Union comprises 15–25 villages, and the pourashava comprises 5–15 wards. Most of the villages have a primary school, and a few primary schools in each pourashava. Narail is a typical rural district located 150 km away from the capital city of Bangladesh. There are more than 220 primary and secondary schools and madrasahs (Islamic educational institutes) with an estimated number of primary school children, 15657, and about 11000 children in high schools and Madarshas combined [29]. Data was collected from all students

enrolled in all primary schools on the day of data collection. Approximately 15% of students were absent on the day of data collection and were out of this research.

## Sample size and statistical power

Prior data indicate that the prevalence of blind children was 0.52% in children aged 1–15 years [30]. The study was at least 90% powered to show a prevalence of 0.52% with a 95% confidence interval of 0.005 percentage points above or below the prevalence. Thus, the required sample size would be 11298 children at a significance level 0.05. Allowing for a 20% non-response rate, which includes missing or incomplete responses, the sample size would be 13557 or approximately 14000.

## Sampling frame and screening strategy

Narail Upazila consists of 13 rural Unions and the urban Pourasava. We have covered screening for vision from all primary schools. Before screening any schools, schoolteachers and the community leaders were informed one week before attending the school to increase students' attendance. One ophthalmic assistant from a local non-government organization (NGO), the Organization for Rural Community Development (ORCD), and another casual employee with an undergraduate degree went to schools for screening. The data collectors, upon arrival and following up on the ethics requirements, including greetings and disseminating the study objectives, the data collectors and two trained schoolteachers collected data using a self-reported questionnaire and a Snell chart for measuring spherical equivalent. The self-reported questionnaire contained parents' socio-demographic status, previous history of eye examination, lifestyle factors, including watching television or using a mobile phone, and ocular conditions, including problems seeing blackboards or using spectacles for any ocular conditions.

## Outcome variables

**Lifestyle and behavioral factors:** Vitamin A taken, with a response of yes or no; Watching TV, with a response of yes or no; the number of hours watching TV, with a response of one hour or more; use of a mobile, with responses of yes or no and the number of hours using a mobile phone, with responses of one hour or more; reading books regularly, with a response of any number of hours between one and six hours.

 **Ocular problems and eye examinations:** problems in seeing blackboard or distance people with responses yes or no, experience headache with responses yes or no, had eye examinations with responses yes or no, use spectacles with responses yes or no, and experience of any ocular trauma with responses yes or no.

## Exposure variables

The exposure variables included children's year of enrolment, gender (boys and girls), living area (urban and rural), housing condition (classified as kacha, which is without any concrete slabs or brick, semi-paka which are made of tin roof, concrete slab and brick wall, and paka which is a building with formal concrete roof), father and mother's education– categorized as no formal education, primary education (grade 1–5), secondary education (grade 6–10), secondary or higher secondary school certificates (national exam after grade 10 and grade 12 with successful outcomes), and Bachelor or Master, and father and mother's occupation. The occupation was categorized as housewives, self-managed businesses, laborers, which included digging soils, pulling rickshaws or any laborious work, and employees who included government and non-government employees; and socioeconomic status (SES)- categorized as very poor who had insufficient funds most of the time in the year before data collection, poor who had adequate funds some of the time, middle class who had neither sufficient funds nor deficit and wealthy who had adequate funds most of the time. The SES was assessed according to Cheng et al.[31].

## Ethics approval

We conducted a vision screening program with the additional objective of reporting scientific evidence, and thus, the study followed the tenets of the Declaration of Helsinki. Human Ethics Approval was received from the Swinburne University of Technology Human Ethics Committee (SHR Project R/2019/017). We obtained written consent from the schoolteachers and the participants. Participants were informed of their right to withdraw from the study at any stage if they wished or directly not to participate if they intended.

## Statistical analysis

Children's characteristics, including a year of enrolment, gender, parental education, occupation, and socioeconomic status, were presented using descriptive statistics with numbers and percentages. We performed bivariate analyses to test the hypotheses that the factors, including gender, parent's education and occupation, living status, and housing condition, were not associated with lifestyle and behavioral characteristics, including Vitamin A taken, watching TV and mobile use, and seeing problems, use of spectacles and history of eye examination.

We also performed bivariate analyses to test the hypotheses that the lifestyle and behavioral factors were not associated with ocular problems, including problems in seeing the blackboard, use of spectacles, or history of the eye examination. Lifestyle or behavioral factors that were significantly associated with ocular problems were determined by chi-square tests, and a binary logistic regression technique was used to estimate the odds ratio (OR) with 95% confidence intervals (CI). The reference categories included no watching television for watching TV and no use of mobile phones for mobile phone use. The reference groups are shown in the results section. Statistical software SPSS (SPSS Inc, version 27) was used for analyses. Any results were statistically significant with less than or equal to a 0.05 significance level.

## Results

Table 1 shows the characteristics of the participants. The majority of participants are between 7–11 years old, with the most dominant age group being 7 years old (22.9%). The sample is consisting of 7004 girls (52.5%) and 6337 boys (47.5%). Most of the participants, 93.3%, reside in rural areas, with only 6.7% residing in urban areas, indicating a highly rural sample. Most participants reside in semi-paka houses (57.9%), followed by those who reside in Kacha houses (24.9%). A smaller proportion, 17.2%, reside in Paka houses, indicating a difference in housing conditions.

A majority of the fathers have secondary education (57.3%), with lower numbers having primary education (12.1%) or higher education (graduate or higher at 8.0%). Notably, 1.8% of the fathers have no education. A majority of the mothers have secondary education (60.2%), with the second most common category being primary education (23.1%). A lower percentage have graduate education (2.2%).

The majority of the fathers work in agriculture (35.2%), followed by daily labor (25.4%) and business (17.2%). Some are unemployed (0.5%) or unable to work (0.5%).

Nearly all the mothers, 96.4%, are homemakers or house duties, and only a small number is engaged in paid work. The remaining, 3.6%, are in daily labor (0.5%) or business (0.1%). The majority of the respondents belong to the middle-class group (69.3%), followed by the poor (16.5%) and the rich (12.7%). Few are very poor (1.6%), indicating a predominantly middle-income sample.

Table 2 presents data on behavioral factors such as TV watching, mobile phone usage, and reading books, along with self-reported ocular problems and ocular trauma and their associated characteristics, including children's age, gender, and parents' sociodemographic factors. The main statistical approach used in this table is the Chi-square test. P for trend tests examine whether there is a significant trend across ordered categories like grade levels and socio-economic status.

The coverage of Vitamin A taken was 98.8%. Of the children, 58.7% watched TV, 58.5% used mobile phones, 17.6% used more than one hour, and 25.1% read books 3–6 hours a day. About 60% of children watched TV or computers. The

**Table 1. Descriptive information with number and percentage of children, and the socio-demographic conditions of the children's family.**

| | | Number | Percentage |
|---|---|---|---|
| Age in years | 7 | 3051 | 22.9 |
| | 8 | 2731 | 20.5 |
| | 9 | 2625 | 19.7 |
| | 10 | 2473 | 18.5 |
| | 11 | 2397 | 18.0 |
| | 12 | 21 | 0.2 |
| | 13 | 20 | 0.1 |
| | 14 | 23 | 0.2 |
| Gender | Girls | 7004 | 52.5 |
| | Boys | 6337 | 47.5 |
| Living Area | Urban Area | 900 | 6.7 |
| | Rural Area | 12441 | 93.3 |
| Housing condition | Kacha | 3318 | 24.9 |
| | Semi Paka | 7722 | 57.9 |
| | Paka | 2301 | 17.2 |
| Father's Education | Missing | 62 | 0.5 |
| | No Formal Education | 243 | 1.8 |
| | Primary Education | 1612 | 12.1 |
| | Secondary (below SSC) | 7644 | 57.3 |
| | SSC or HSC Pass | 2717 | 20.4 |
| | Graduate or More | 1063 | 8.0 |
| Mother's Education | Missing | 62 | 0.5 |
| | No Formal Education | 289 | 2.2 |
| | Primary Education | 3082 | 23.1 |
| | Secondary (below SSC) | 8036 | 60.2 |
| | SSC or HSC Pass | 1582 | 11.9 |
| | Graduate or more | 290 | 2.2 |
| Father's Occupation | Govt or Private Job | 1951 | 14.6 |
| | Business | 2299 | 17.2 |
| | Farmer or Agriculture related | 4700 | 35.2 |
| | Daily Labour | 3385 | 25.4 |
| | Unemployed | 66 | 0.5 |
| | Can't work | 72 | 0.5 |
| | Home maker or House Duties | 42 | 0.3 |
| | Others | 826 | 6.2 |
| Mother's Occupation | Govt or Private Job | 248 | 1.9 |
| | Business | 16 | 0.1 |
| | Farmer or Agriculture Related | 41 | 0.3 |
| | Daily Labour | 62 | 0.5 |
| | Unemployed | 69 | 0.5 |
| | Can't Work | 27 | 0.2 |
| | Homemaker or House Duties | 12861 | 96.4 |
| | Others | 17 | 0.1 |
| Socio-Economic Status | Very poor, insufficient funds most of the time | 209 | 1.6 |
| | Poor, adequate funds some of the time | 2198 | 16.5 |
| | Middle class, neither sufficient funds nor deficit | 9242 | 69.3 |
| | Wealthy, adequate funds most of the time | 1692 | 12.7 |

**Table 2. The association of sociodemographic factors with lifestyle and behavioral characteristics and self-reported ocular problems in the 13341 children reported using Chi-square tests.**

| Characteristics | No at risk | Lifestyle and behavioural factors | | | | | | Self-reported ocular problems, eye examination, spectacle use and family history of eye disease | | | | |
|---|---|---|---|---|---|---|---|---|---|---|---|---|
| | | Vitamin A | Watch TV | Watch TV>1 hr | Watch mobile | use mobile>1 hr | *Read book, 3–6 hrs* | Seeing problem, n=99 | Headach, n=41 | Eye examination, n=59 | Use spectacles, n=32 | Ocular trauma, n=34 |
| | | % | % | % | % | % | % | % | % | % | % | % |
| Total | 13341 | 98.8 | 58.7 | 4.4 | 58.5 | 17.6 | 25.1 | 0.7 | 0.3 | 0.4 | 0.2 | 0.3 |
| Enrolment Grades | | | | | | | | | | | | |
| Grade one | 3051 | 98.9 | **55.1** | **3.5** | **56.5** | **14.5** | **1.7** | 0.8 | 0.2 | 0.6 | 0.2 | 0.3 |
| Grade Two | 2731 | 98.8 | **57.9** | **3.8** | **58.4** | **17.7** | **7.1** | 0.5 | 0.2 | 0.8 | 0.1 | 0.1 |
| Grade Three | 2625 | 98.9 | **59.2** | **5.3** | **57.8** | **17.7** | **26.7** | 0.6 | 0.2 | 4.8 | 0.3 | 0.2 |
| Grade Four | 2473 | 99.1 | **61.9** | **4.6** | **58.9** | **17.7** | **43.1** | 0.7 | 0.4 | 0.0 | 0.4 | 0.2 |
| Grade Five | 2397 | 98.4 | **61.0** | **6.2** | **61.7** | **21.6** | **53.8** | 1.1 | 0.6 | 0.0 | 0.3 | 0.5 |
| P for trend* | | 0.16 | <0.001 | <0.001 | <0.001 | <0.001 | <0.001 | 0.43 | 0.18 | 0.002 | 0.52 | 0.14 |
| Gender | | | | | | | | | | | | |
| Girls | 7004 | 98.6 | 58.4 | 4.5 | 58.5 | 16.7 | 26.3 | 0.7 | 0.3 | 0.4 | 0.2 | 0.2 |
| Boys | 6337 | 99.0 | 59.1 | 4.8 | 58.6 | 18.7 | 24.0 | 0.8 | 0.4 | 0.3 | 0.3 | 0.3 |
| P* | | 0.02 | 0.41 | 0.52 | 0.87 | 0.02 | 0.003 | 0.42 | 0.43 | 0.02 | 0.52 | 0.77 |
| Living area | | | | | | | | | | | | |
| Urban Area | 900 | 99.9 | **67.1** | 4.0 | **69.6** | 15.3 | 25.1 | **2.2** | **1.2** | 0.5 | 0.3 | 0.1 |
| Rural Area | 12441 | 98.7 | **58.1** | 4.7 | **57.8** | 17.9 | 25.2 | **0.6** | **0.3** | 0.6 | 0.2 | 0.3 |
| P* | | 0.008 | <0.001 | 0.42 | <0.001 | 0.11 | 0.96 | <0.001 | <0.001 | 0.29 | 0.55 | 0.38 |
| Housing condition | | | | | | | | | | | | |
| Kacha | 3318 | **99.9** | **42.3** | **1.2** | **38.3** | **6.1** | **20.6** | 0.6 | 0.2 | 0.8 | 0.2 | 0.1 |
| Semi Paka | 7722 | **98.0** | **62.1** | **6.2** | **62.5** | **24.3** | **22.0** | 0.6 | 0.2 | 0.5 | 0.2 | 0.3 |
| Paka | 2301 | **99.8** | **71.1** | **3.1** | **74.5** | **7.8** | **42.5** | 1.3 | 0.6 | 0.3 | 0.3 | 0.1 |
| P* | | <0.001 | <0.001 | <0.001 | <0.001 | <0.001 | <0.001 | 0.006 | 0.02 | 0.20 | 0.77 | 0.04 |
| Father's education | | | | | | | | | | | | |
| No Formal Education | 243 | **96.7** | **51.9** | **13.5** | **51.4** | **29.6** | **22.3** | 0.8 | 0.0 | 0.6 | 0.4 | 0.0 |
| Primary Education | 1612 | **97.9** | **52.0** | **5.5** | **49.3** | **31.8** | **13.7** | 0.7 | 0.6 | 0.8 | 0.3 | 0.7 |
| Secondary school | 7644 | **98.8** | **54.3** | **4.2** | **54.3** | **17.1** | **23.3** | 0.7 | 0.2 | 0.7 | 0.2 | 0.2 |
| SSC or HSC Pass | 2717 | **99.4** | **70.6** | **3.8** | **70.0** | **11.8** | **31.8** | 0.7 | 0.3 | 0.5 | 0.3 | 0.1 |
| Graduate or more | 1063 | **99.4** | **74.0** | **6.6** | **77.5** | **18.3** | **40.7** | 1.0 | 0.5 | 0.4 | 0.6 | 0.4 |
| P for trend* | | <0.001 | <0.001 | <0.001 | <0.001 | <0.001 | <0.001 | 0.85 | 0.19 | 0.22 | 0.10 | 0.06 |
| Mother's education | | | | | | | | | | | | |
| No Formal Education | 289 | **97.6** | **49.5** | **5.6** | **51.9** | **40.0** | **16.6** | **0.3** | **0.3** | 0.4 | 0.3 | 1.0 |
| Primary Education | 3082 | **97.9** | **52.8** | **5.5** | **50.1** | **26.1** | **15.4** | **0.7** | **0.4** | 1.4 | 0.2 | 0.3 |
| Secondary school | 8036 | **99.0** | **58.6** | **4.3** | **58.5** | **15.1** | **26.4** | **0.7** | **0.2** | 0.6 | 0.2 | 0.2 |
| SSC or HSC Pass | 1582 | **99.7** | **71.4** | **4.3** | **74.0** | **13.8** | **35.8** | **0.8** | **0.5** | 0.6 | 0.4 | 0.2 |
| Graduate or more | 290 | **100.0** | **73.8** | **5.6** | **80.3** | **17.2** | **46.6** | **3.1** | **1.4** | 0.3 | 0.7 | 0.3 |
| P for trend* | | <0.001 | <0.001 | 0.34 | <0.001 | <0.001 | <0.001 | <0.001 | 0.004 | 0.12 | 0.36 | 0.08 |
| Father's occupation | | | | | | | | | | | | |

*(Continued)*

**Table 2.** (Continued)

| Characteristics | No at risk | Lifestyle and behavioural factors | | | | | | Self-reported ocular problems, eye examination, spectacle use and family history of eye disease | | | | |
|---|---|---|---|---|---|---|---|---|---|---|---|---|
| | | Vitamin A | Watch TV | Watch TV>1 hr | Watch mobile | use mobile>1 hr | Read book, 3–6 hrs | Seeing problem, n=99 | Headach, n=41 | Eye examination, n=59 | Use spectacles, n=32 | Ocular trauma, n=34 |
| | | % | % | % | % | % | % | % | % | % | % | % |
| Govt or Private Job | 1951 | **99.3** | 72.5 | 5.6 | 75.2 | 15.5 | 35.8 | 0.9 | 0.4 | 0.4 | 0.3 | 0.3 |
| Business | 2299 | **98.9** | 63.4 | 4.6 | 64.5 | 17.7 | 27.8 | 0.9 | 0.5 | 0.0 | 0.3 | 0.1 |
| Farmer or Agriculture | 4700 | **98.1** | 56.1 | 5.2 | 55.9 | 21.5 | 22.3 | 0.6 | 0.2 | 0.0 | 0.2 | 0.3 |
| Daily Labour | 3385 | **99.4** | 48.2 | 3.4 | 44.7 | 15.4 | 20.9 | 0.7 | 0.3 | 2.4 | 0.1 | 0.2 |
| Unemployed | 66 | **100.0** | 62.1 | 2.4 | 66.7 | 4.5 | 18.5 | 0.0 | 0.0 | 0.5 | 0.0 | 0.0 |
| Can't work | 72 | **100.0** | 40.3 | 3.4 | 37.5 | 11.1 | 26.4 | 0.0 | 0.0 | 1.6 | 0.0 | 0.0 |
| Home maker | 42 | **100.0** | 61.9 | 3.8 | 52.4 | 9.1 | 9.5 | 0.0 | 0.0 | 0.0 | 2.4 | 2.4 |
| Others | 826 | **98.5** | 72.5 | 3.8 | 76.2 | 13.7 | 28.3 | 0.7 | 0.1 | 0.0 | 0.2 | 0.2 |
| P* | | **<0.001** | **<0.001** | 0.13 | **<0.001** | **<0.001** | **<0.001** | 0.85 | 0.53 | 0.28 | 0.10 | 0.11 |
| Mother's occupation | | | | | | | | | | | | |
| Govt or Private Job | 248 | 100.0 | **78.2** | **5.7** | 84.7 | 14.3 | **44.4** | 2.4 | 1.2 | **1.6** | 0.8 | 0.4 |
| Business | 16 | 100.0 | **62.5** | **0.0** | 68.8 | 0.0 | **31.3** | 0.0 | 0.0 | **4.3** | 0.0 | 0.0 |
| Farming/ Agriculture | 41 | 95.1 | **58.5** | **0.0** | 58.5 | 4.2 | **22.0** | 0.0 | 0.0 | **0.0** | 0.0 | 2.4 |
| Daily Labour | 62 | 100.0 | **51.6** | **3.1** | 41.9 | 7.7 | **14.5** | 1.6 | 0.0 | **0.4** | 1.6 | 0.0 |
| Unemployed | 69 | 98.6 | **18.8** | **30.8** | 21.7 | 14.3 | **18.4** | 0.0 | 0.0 | **0.0** | 0.0 | 0.0 |
| Can't Work | 27 | 100.0 | **29.6** | **12.5** | 44.4 | 41.7 | **23.1** | 3.7 | 0.0 | **1.4** | 0.0 | 0.0 |
| Home Maker | 12861 | 98.8 | **58.7** | **4.6** | 58.4 | 17.9 | **24.9** | 0.7 | 0.3 | **0.6** | 0.2 | 0.2 |
| Others | 17 | 88.2 | **52.9** | **0.0** | 41.2 | 0.0 | **29.4** | 0.0 | 0.0 | **0.3** | 0.0 | 0.0 |
| P* | | 0.03 | **<0.001** | 0.001 | **<0.001** | 0.03 | **<0.001** | 0.04 | 0.40 | **<0.001** | 0.27 | 0.30 |
| Socioeconomic status | | | | | | | | | | | | |
| Very poor, insufficient funds most of the time | 209 | **100.0** | 84.2 | 5.1 | 86.6 | 4.4 | 42.6 | **2.9** | 0.5 | 0.6 | 0.5 | 0.0 |
| Poor, adequate funds some of the time | 2198 | **99.7** | 75.6 | 2.4 | 76.4 | 7.1 | 37.3 | **1.2** | 0.6 | 0.6 | 0.5 | 0.1 |
| Middle class, neither sufficient funds nor deficit | 9242 | **98.9** | 55.2 | 5.3 | 54.7 | 21.3 | 21.8 | **0.6** | 0.2 | 0.8 | 0.2 | 0.2 |
| Wealthy, adequate funds most of the time | 1692 | **97.0** | 53.3 | 5.2 | 53.0 | 19.5 | 26.0 | **0.8** | 0.3 | 4.8 | 0.1 | 0.5 |
| P for trend* | | **<0.001** | **<0.001** | **<0.001** | **<0.001** | **<0.001** | **<0.001** | **0.002** | 0.05 | 0.19 | 0.02 | 0.007 |

*P for Chi-square tests; Bonferoni corrected p = (0.05/9)=0.005.

chi-square test revealed that gender was not associated with watching TV (boys 59.1% boys vs. 58.4% girls, p=0.41). However, watching TV was significantly higher in urban areas (67.1% urban vs. 58.1% rural areas, p<0.001) and who owned a paka house (71.1% in paka house vs. 42.3% in kacha houses, p<0.001). The proportion of watching TV or using mobile phones was higher among children whose parents had higher education, had government or private jobs, or those with higher socio-economic status. The proportion of watching TV or using mobile phones for more hours was higher among children whose parents had no education or primary education. Almost every student reads books regularly, but the percentage of more hours spent reading was higher for children in higher grades, in paka houses, and parents with higher education. For example, the proportion of children in grade 1 who read books 3–6 hours per day was 1.7%, and those in grade 5 was 53.8%. Test for the trend option in the Chi-square test revealed a significant trend as they went to the higher grades, p<0.001 for trend. Ninety-nine (0.7%) children reported that they had problems seeing the blackboard or distance people, with a significant difference between urban (2.2%) and rural areas (0.6%), p<0.001, and between living in paka houses (1.3%) and kacha or semi-paka houses (0.6), p=0.006. Mothers' higher education was associated with more hours of reading books (16.6% in mothers with no formal education vs. 46.6% in graduate mothers, p<0.001) and a higher proportion of seeing problems (0.3% in mothers with no formal education vs. 3.1% in graduate mothers, p<0.001) and a higher proportion of headache (0.3% in mothers with no formal education vs. 1.4% in graduate mothers, p=0.004). Only 59 (0.4%) students had a history of previous eye examination, 32 (0.2%) used spectacle, and four had previous eye surgery.

Table 3 includes Chi-square tests and p-values that help interpret the relationships between different lifestyles and behavioral factors with self-reported ocular conditions, history of eye examination, and spectacle use. Watching TV or

**Table 3. The association of lifestyle and behavioral factors with self-reported ocular problems, eye examinations, and spectacles use in the total sample of 13341 children reported using Chi-square tests.**

| Lifestyle and behavioural factors | No at risk | Seeing problem, n=99 (0.7%) | | Headache, n=41 (0.3%) | | Eye examination, n=59 (0.4%) | | Spectacles, n=32 (0.2%) | | Any ocular trauma, n=34 (0.3%) | |
|---|---|---|---|---|---|---|---|---|---|---|---|
| | N | n | % | N | % | N | % | n | % | N | % |
| Watch Computers or TV frequently | | | | | | | | | | | |
| Yes | 7837 | 69 | 0.9 | 26 | 0.3 | 40 | 0.5 | 18 | 0.2 | 22 | 0.3 |
| No | 5504 | 30 | 0.5 | 15 | 0.3 | 19 | 0.3 | 14 | 0.3 | 12 | 0.2 |
| P* | | | 0.03 | | 0.56 | | 0.16 | | 0.77 | | 0.48 |
| Watch TV hrs per day | | | | | | | | | | | |
| one hour | 7473 | 58 | **0.8** | 24 | 0.3 | 39 | 0.5 | 16 | 0.2 | 19 | 0.3 |
| >1 hour | 364 | 11 | **3.0** | 2 | 0.6 | 1 | 0.3 | 2 | 0.5 | 3 | 0.8 |
| P* | | | **<0.001** | | 0.47 | | 0.52 | | 0.19 | | 0.05 |
| Use mobile frequently | | | | | | | | | | | |
| Yes | 7811 | 63 | 0.8 | 24 | 0.3 | 32 | 0.4 | 15 | 0.2 | 23 | 0.3 |
| No | 5530 | 36 | 0.7 | 17 | 0.3 | 27 | 0.5 | 17 | 0.3 | 11 | 0.2 |
| P* | | | 0.30 | | 0.99 | | 0.50 | | 0.18 | | 0.28 |
| Use mobile (hrs per day) | | | | | | | | | | | |
| 1 hour | 6430 | 47 | 0.7 | 22 | 0.3 | 19 | **0.3** | 9 | 0.1 | 8 | 0.1 |
| >1 hours | 1380 | 16 | 1.2 | 2 | 0.1 | 13 | **0.9** | 6 | 0.4 | 15 | 1.1 |
| P* | | | 0.11 | | 0.22 | | **<0.001** | | 0.02 | | **<0.001** |
| Read books (hrs per day) | | | | | | | | | | | |
| 1–2 hours | 9945 | 59 | **0.6** | 23 | **0.2** | 38 | 0.4 | 19 | 0.2 | 20 | 0.2 |
| 3–6 hours | 3351 | 40 | **1.2** | 18 | **0.5** | 20 | 0.6 | 13 | 0.4 | 14 | 0.4 |
| P* | | | **<0.001** | | **0.006** | | 0.10 | | 0.04 | | |

*P for the test of association using Chi-square tests; **Bonferoni corrected p = (0.05/6)=0.008.**

a computer or using a mobile phone was not associated with any ocular condition, but more hours of watching TV or using mobiles or reading books were associated with a higher proportion of seeing problems or any ocular trauma. Only five children did not take vitamin A, one of whom had problems seeing the blackboard, had eye examinations, and used spectacles.

Table 4 presents data on behavioral factors and their associations with ocular problems and ocular trauma. The main statistical approach used in this table is the relative risk (RR), which assesses the strength of the association. The statistical analysis, based on relative risk, shows that longer durations of TV watching and mobile usage are significantly associated with an increased likelihood of experiencing ocular problems and ocular trauma. Reading books for 3–6 hours daily is also linked to a higher likelihood of seeing problems. The findings demonstrate the potential role of screen time (mobile and TV) on eye health. Compared to children who did not watch TV or computers, children who watched TV or computers for one-hour, relative risk (RR) (95% CI) 4.16 (2.15, 8.07) or more than one hour RR (95% CI) 5.33 (2.62, 10.8) were more likely to have seeing problems. Compared to children who did not use mobile phone, those who used it for one hour RR (95% CI) 4.82 (2.17, 10.7) or more than one hour RR (95% CI) 7.30 (3.06, 17.4) were more likely to have any ocular trauma. Reading more hours was also associated with a higher proportion of seeing problems.

## Discussion

Ocular problems have a significant impact on daily life, including poor cognitive function and learning deficiency. Understanding the ocular problems and practice in the detection of vision impairment and the factors that potentially cause vision impairment are primary approaches for preventing vision impairment [14,15]. The significant findings from this study include: (1) Almost 99% of the children had taken Vitamin A, (2) About 60% of the children watched TV or computer or mobile screens regularly and this percentage was higher in urban areas, (3) a small number of children had previous eye examination, (4) only 0.2% of children used spectacles, (5) those who watched TV or used mobile more than one hour daily had problems seeing blackboard or distance people or had any ocular trauma than children who did not watch TV or did not use mobile phones.

Results from this study indicate that about 60% of children are regularly exposed to screens, with a higher prevalence in urban areas, consistent with the research findings. Studies have consistently stated that urban children usually have greater access to electronic devices, leading to increased screen time than their rural counterparts. For instance, one of the national studies in China documented that excessive recreational screen time (RST) among urban children and

**Table 4. The associations of lifestyle and behavioral factors with self-reported ocular problems in the total sample of 13341 children.**

| Behavioural factors | No at risk | Seeing problem, n=99 (0.7%) | | Any ocular trauma, n=34 (0.3%) | |
|---|---|---|---|---|---|
| Watch TV hrs per day | N | n (%) | OR (95% CI)* | n (%) | OR (95% CI)* |
| Do not watch TV | 5504 | 30 (0.5) | Ref. | 12 (0.2) | Ref. |
| one hour | 7473 | 58 (0.8) | **4.16 (2.15, 8.07)** | 19 (0.3) | 2.71 (0.79, 9.02) |
| >1 hour | 364 | 11 (3.0) | **5.33 (2.62, 10.8)** | 3 (0.8) | 3.29 (0.92, 11.9) |
| Use mobile (hrs per day) | | | | | |
| Do not use mobile | 5531 | 36 (0.7) | Ref. | 11 (0.2) | **Ref** |
| 1 hour | 6430 | 47 (0.7) | 1.66 (0.91, 3.03) | 8 (0.1) | **4.82 (2.17, 10.7)** |
| >1 hours | 1380 | 16 (1.2) | **1.76 (1.0, 3.16)** | 15 (1.1) | **7.30 (3.06, 17.4)** |
| Read books (hrs per day) | | | | | |
| 1–2 hours | 9945 | 59 (0.6) | Ref | 20 (0.2) | Ref. |
| 3–6 hours | 3351 | 40 (1.2) | **2.17 (1.33, 3.56)** | 14 (0.4) | 2.05 (0.90, 4.67) |

*Odds ratio (95% confidence interval) adjusted for year levels, gender, living area, living condition, fathers' education, and parent's financial conditions.

adolescents was higher than their rural counterparts. This finding has been attributed to the high availability of electronic devices and the varying lifestyle patterns in urban areas [32]. A recent study reported that teens living in metropolitan areas had higher odds of having four or more hours of screen time on a weekday than their non-metropolitan counterparts [33]. In this regard, a study done among 2–6-year-old children in western India reported an average screen time of 2.7 hours, which was higher among young children [34].

Our study shows that more urban children had more difficulty seeing the blackboard than rural children. A cross-sectional study in India showed a refractive error prevalence rate of 7.61% among urban school children, while it was 5.21% among rural school children. Such difference can be because children in urban settings might run a greater risk of myopia due to some aspects of urban lifestyle, like increased use of computers and mobile phones and lesser time utilization outdoors [35]. Our study reported that children who watched TV for at least one hour were more likely to have seeing problems or ocular trauma than those who did not. These findings demonstrate the potential consequence of long-term screen exposure on children's visual health. This increased risk of more hours of TV watching stresses the need to monitor screen time to prevent children's vision problems. These risks could, therefore, be reduced by promoting outdoor activities, reducing screen exposure, and promoting eye examination, thus improving outcomes in eye health. The results of this study concur with global evidence on the strong association between prolonged screen time and vision problems in children. Likewise, a study by Holden et al. [36] reported a strong correlation between increased screen time and the prevalence of myopia among children. The study also noted that prolonged near-work activities, including watching TV, were significant risk factors for vision problems. In a comparative study between urban and rural school-going children regarding refractive errors, myopia showed an urban preponderance of 52% compared with other parts where a rural preponderance of hyperopia was most seen. The present urban predominance of myopia may contribute to the visual problem in viewing distant objects like blackboards [37].

Our study reports an association between mothers' higher education and a higher proportion of ocular problems. Education has been reported to act as a proxy for the family's socioeconomic status and geographic area of residence. Thus, the correlation between education and ocular problems may be spurious. Educated mothers can identify health problems in children more quickly easily and seek necessary care. Based on data from 22 developing countries, one study reported a correlation between maternal education and markers of child health, which may imply that educated mothers are more concerned with their children's health conditions, including eye defects [38]. Higher reporting of visual problems among children of literate mothers could also be associated with better socioeconomic status, health-seeking behavior, and utilization of health services [39, 40]. Although a higher proportion of ocular problems in our study is self-reported, the children of educated mothers could be more aware of children's conditions than those of uneducated mothers. Our study reported that only a few children had proper eye examinations. Low rates of pediatric eye examinations due to limited healthcare infrastructure and awareness are quite common in low and middle-income countries. For example, in a door-to-door screening of 32857 children aged 5–18 years in an urban slum of Delhi, it was reported that 2.8% of children had previous eye examinations before data collection in 2020 [41]. Yet similar results have been reported among a sub-Saharan population where school-aged children's access to eye care facilities is very low [42]. Most pediatric ophthalmologists and specialized facilities are based in urban areas, meaning many rural areas have limited availability [43].

Our study showed a strong association of ocular trauma in children with the use of a mobile phone. Although few studies have directly investigated a possible link between mobile phone use and ocular trauma, the literature generally supports this association in the context of screen time contributing to disorders of the eyes among children [44, 45]. Some of the studies showed that extensive smartphone uses leads to significant ocular discomfort and reduced tear stability in children. Studies report that even one hour of continuous screen exposure can induce visual strain in children aged 6–15 [46, 47]. The literature, however, remains scanty regarding direct evidence on specific associations between the use of a mobile phone and ocular trauma. For example, a study from the UK listed various causes of eye injuries but did not identify mobile phone use as a significant factor [48].

## Implications and recommendations

Several implications and recommendations can be made based on the findings from this large study. (1) The current study highlights that limiting children's screen time, maintaining distance between the eye and the screen should be at least 50−63 cm, providing regular breaks using the 20-20-20 rule defined as taking regular breaks every 20 minutes by looking at object 20 feet away for 20 seconds, encouraging outdoor activities, and incorporating regular vision screenings into pediatric care, can help mitigate these risks and support eye health [49]. Considering the association of more screen time with a higher proportion of visual problems, parents, educators, and policymakers are in the best position to ensure that the time children spend in front of a screen is tracked and limited, especially in urban settings where the risk is higher. Increasing awareness can be done by implementing structured guidelines and promoting other recreational activities to reduce the possibility of any adverse effects of excessive screen exposure [50]. (2) Though there is limited direct evidence linking mobile phone use with ocular trauma in children, the association of prolonged screen time with various ocular discomforts and impairments is well-documented [51]. Preventive measures can help mitigate these risks. (3) The higher prevalence of reported visual difficulties among urban children and children of educated mothers may indicate actual differences in visual health, health awareness, and availability of diagnostic services. It, therefore, calls for Regular Vision Screenings and routine eye checkups at schools, especially in rural settings, to uncover and attend to unreported visual incapacities. Research has shown that school-based vision programs can improve students' reading scores over one year, thus showing the academic benefits of addressing visual impairments in school settings [52]. (4) The study adds value to Parental Education Programs by educating them about eye health and the importance of early detection of vision problems regardless of their educational background. (5) The findings of higher risk of vision problems for more than one hour of exposure to TV or mobile screens have implications for public health, education, parental practices, and policymaking.

Schools can play a pivotal role by integrating screen time policies, including lessons on visual hygiene and training teachers to identify early signs of vision problems. These findings show that parents must monitor and limit screen exposure while encouraging outdoor activities to establish healthier visual habits. Additionally, governments can address such concerns through policies regulating children's screen time in schools and subsidizing vision care services for under-resourced populations, such as vitamin A supplements, to make eye care services successful [51]. An integrated, multi-faceted strategy is required to respond to the critical gap the screening program reveals in pediatric eye care. First, it has to be integrated into school health programs: School health initiatives should include mandatory vision screening to detect visual impairment early enough. Since teachers are among the first observers of the children's behavior and performance, the teachers must be trained to recognize early signs of poor vision and refer students in time for proper treatment. Community-based campaigns must highlight that eye testing is necessary for parents and guardians. Long-term benefits of regular vision checkups can be made known to the families during the educational programs to instil a culture of prevention. Resource allocation should be channelled toward mobile vision clinics and telemedicine solutions, especially in far-reaching or less privileged areas, to bridge the gap in service access. Such initiatives can bring critical eye care services right to the doorsteps of communities that otherwise lack access. In indigent families, subsidised vision care ensures that financial constraints do not hinder children from receiving the necessary care. The efforts will be institutionalized through a strong policy framework. By clearly setting guidelines, national policies must advocate for early and periodic pediatric eye examinations. This clear guideline would be a sustainable safeguard for children's vision, health, and overall development.

## Strengths and limitations

Our study has several strengths: First, it includes a prominent community, especially schoolteachers who are well accepted and trusted in the community. The participation rate in the study was high, which can be due to the well-established trust and respect among students, teachers and the students' guardians. Higher participation in the vision test and face-to-face interview is consistent with previous studies where vision was screened by community health workers (CHW) [53]. Second, the study is based on a large sample and face-to-face data collection. The potential drawback of

our study is that it is a report from a single-occasion collection of data from a single location. Although the study location is similar to other districts in Bangladesh, the study would need to be repeated in a random sample of different districts for the results to represent a national perspective truly. Data are self-reported, which does not exclude the possibility of reporting errors. Especially, reliance on parental questionnaires such as mothers' education and questions related to their daily living, such as watching TV for specific hours, is likely that the answers did not accurately reflect children's viewpoints. Children's reports were not validated, though teachers checked the accuracy to minimize the reporting error as much as possible. Another limitation was that the percentage of self-reported ocular problems was small, such as headache 41 (0.3%) and ocular trauma 34 (0.3%). Factors associated with the small number suffer from the insufficient statistical power, and the conclusion may not reflect the truth. However, the total sample size was already large enough. The study has raised concerns and needs further study with greater details.

We attempted to collect data from all students but could not repeat the data collection process to bring them absent from the school on the screening day. However, the attendance rate was more than 85%.

## Conclusion

The current study shows that the proportion of children taking vitamin A, a government-initiated program, was very high. Although the proportion of children with vision problems, headaches, or any ocular trauma is not high, these incidents were higher in urban areas and who spent more time watching TV or mobile screens. The associations of mothers' higher education with a higher proportion of vision problems can be considered a proxy for the socioeconomic status and urban areas. Due to the skill shortage, especially of ophthalmologists in rural areas, engaging trained schoolteachers for school-based screening purposes can be a feasible and cost-effective approach to vision care at community levels. Factors associated with the small number of self-reported visual problems suffer from insufficient statistical power, and the conclusion may not reflect the truth. The study has raised concerns and needs further study with greater details.

## Supporting information

**S1 Data. Lifestyle factor_PLOS One.**
(XLSX)

## Acknowledgments

We want to extend our deepest appreciation to all who made this study a success. We want to thank Md Sadequl Islam, the executive director of the ORCD, for his tremendous effort to communicate with all schools, including contacting the administration and the Department of Primary Education. We want to thank Lima Asma, Tarin Sharmin, Jamirul Islam, Mafijur Rahman, Nupur, and Ratul, who participated in the screening program. We are grateful to more than 800 schoolteachers and their staff from 192 schools for their wholehearted cooperation—my thanks to 13341 schoolchildren.

## Author contributions

**Conceptualization:** Fakir M Amirul Islam.

**Data curation:** Fakir M Amirul Islam, Arzan Hosen.

**Formal analysis:** Fakir M Amirul Islam.

**Funding acquisition:** Fakir M Amirul Islam.

**Investigation:** Fakir M Amirul Islam.

**Methodology:** Arzan Hosen.

**Project administration:** Arzan Hosen.

**Writing – original draft:** Fakir M Amirul Islam.

**Writing – review & editing:** Abdullah Al Mahmud.

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
