## [Decision Letter · Decision Letter 0]

PONE-D-25-00919Association of lifestyle and behavioral factors with self-reported visual problems among schoolchildren in rural Bangladesh.PLOS ONE

Dear Dr. Islam,

Thank you for submitting your manuscript to PLOS ONE. After careful consideration, we feel that it has merit but does not fully meet PLOS ONE’s publication criteria as it currently stands. Therefore, we invite you to submit a revised version of the manuscript that addresses the points raised during the review process.

We look forward to receiving your revised manuscript.

Kind regards,

Nick Fogt

Academic Editor

PLOS ONE

Journal Requirements:

Additional Editor Comments (if provided):

Please address in detail the comments from reviewer #2. In particular, please carefully consider the comments related to limitations of the sampling methods, as well as suggestions regarding the statistical analyses. Please clarify the statistical analyses and results.

Reviewers' comments:

Reviewer's Responses to Questions

**Comments to the Author**

1. Is the manuscript technically sound, and do the data support the conclusions?

Reviewer #1: Yes

Reviewer #2: Partly

2. Has the statistical analysis been performed appropriately and rigorously? 

Reviewer #1: Yes

Reviewer #2: N/A

3. Have the authors made all data underlying the findings in their manuscript fully available?

Reviewer #1: Yes

Reviewer #2: No

4. Is the manuscript presented in an intelligible fashion and written in standard English?

Reviewer #1: Yes

Reviewer #2: Yes

5. Review Comments to the Author

Reviewer #1: The article full filled all rules. Appreciate the title, Introduction , Material and method, statistical analysis, Result, discussion and reference. The sample size, study design, table presentation all were appropriately designed.

Reviewer #2: This is a cross-sectional study including a large sample of participants; however, the reduced number of statistical subgroups limits their statements. The authors present the self-reported ocular problems of schoolchildren mainly in grades one to five in the community of Narail, Bangladesh, as well as the lifestyle, behavioral and sociodemographic factors that are associated with these problems. The data presented can inform initiatives to enhance eye health care for school children in Bangladesh by highlighting the current status.

Regarding the study design and data interpretation possible limitations by the self-reported data collection from children, mainly aged 7-11 should be discussed and possible misreporting and inaccuracies should be clearly addressed. Furthermore, the small number of cases within the subgroups should be discussed as possible limitations within the overall large sample size. In some cases the interpretation of results implies causality. Given the study design only associations can be made. Please revise the manuscript accordingly.

Considering the clarity and definitions used throughout the manuscript please ensure to define the term of financial solvency numerically especially by clearly defining its catergories “all/most/some/none of the time”. On page 5 the study aims listed as (i),(ii) and (iii) should be listed correctly and be used throughout the manuscript. Overall sentence structure should be improved, as potentially independent statements should be cleary separated as on Page12: “ Of the total children, 99 (0.7%) reported that they had problems seeing the blackboard or distance people, with a significant difference between urban (2.2%) and rural areas (0.6%), p<0.001, and those who were living in paka houses (1.3%).”.

The statistical reporting within the manuscript should be presented comprehensively and uniform, including the name of the tests used, p-values, and confidence intervals. For example, on page 11, the statement “significantly higher in urban areas (67.1% vs. 58.1%)” requires supporting statistics. Tables should include description for each number featured (table 1), as well as statistical approaches used should be explained for tables presenting statistical calculation results (tables 2-4). The association between mobile phone use and ocular traumas should be presented with caution, given the low number of trauma cases (n=34).

Check the consistency of your reported results in both the text and the tables, as these do not match on page 15. You state “Only five children did not take vitamin A, of whom one (20%) had problems seeing the blackboard, had headaches, had eye examinations, and used spectacles.”, whereas in table 3, you report that no children with headaches took no vitamin A.

To improve specific data interpretation, please clarify your statement on page 18: “Our study reported that for children who watched TV for more than one hour or used a mobile screen for more than one hour, the relative risk was significantly higher than those who did not watch TV or did not spend time on a mobile screen.”.

Furthermore the discussion provides important contextualization of the findings. However, referencing data from 2002 (Murthy et al.) may be outdated. A more recent comparison would strengthen the relevance of the discussion.

Overall the manuscript provides valuable insights into the topic of pediatric eye health in a rural setting; however, it is essential that the text is subjected to several revisions in order to ensure clarity, statistical rigor, and appropriate interpretation of the findings. It is submitted that the aforementioned revisions would result in the manuscript making a meaningful contribution to the field of global eye health.

6. PLOS authors have the option to publish the peer review history of their article (what does this mean? ). If published, this will include your full peer review and any attached files.

**Do you want your identity to be public for this peer review?** For information about this choice, including consent withdrawal, please see our Privacy Policy .

Reviewer #1: **Yes: ** Prof. Khair Ahmed Choudhury

Reviewer #2: No

---

## [Author Response · Author response to Decision Letter 1]

27 May 2025

Reviewer 1 did not have any comments to address. For Reviewer 2, letter is attached.

---

## [Decision Letter · Decision Letter 1]

PONE-D-25-00919R1Association of lifestyle and behavioral factors with self-reported visual problems among schoolchildren in rural Bangladesh.PLOS ONE

Dear Dr. Islam,

Thank you for submitting your manuscript to PLOS ONE. After careful consideration, we feel that it has merit but does not fully meet PLOS ONE’s publication criteria as it currently stands. Therefore, we invite you to submit a revised version of the manuscript that addresses the points raised during the review process.

We look forward to receiving your revised manuscript.

Kind regards,

Nick Fogt

Academic Editor

PLOS ONE

Journal Requirements:

Additional Editor Comments:

Thank you for your thoughtful responses to the reviewer comments. One final thing: could you please bold or highlight those findings with significant p values in Tables 2 and 3. There is obviously a substantial amount of information in these tables, and it would be easier to process them with the suggested highlighting.

Reviewers' comments:

Reviewer's Responses to Questions

**Comments to the Author**

1. If the authors have adequately addressed your comments raised in a previous round of review and you feel that this manuscript is now acceptable for publication, you may indicate that here to bypass the “Comments to the Author” section, enter your conflict of interest statement in the “Confidential to Editor” section, and submit your "Accept" recommendation.

Reviewer #2: All comments have been addressed

2. Is the manuscript technically sound, and do the data support the conclusions?

Reviewer #2: (No Response)

3. Has the statistical analysis been performed appropriately and rigorously? 

Reviewer #2: (No Response)

4. Have the authors made all data underlying the findings in their manuscript fully available?

Reviewer #2: (No Response)

5. Is the manuscript presented in an intelligible fashion and written in standard English?

Reviewer #2: (No Response)

6. Review Comments to the Author

Reviewer #2: (No Response)

7. PLOS authors have the option to publish the peer review history of their article (what does this mean? ). If published, this will include your full peer review and any attached files.

**Do you want your identity to be public for this peer review?** For information about this choice, including consent withdrawal, please see our Privacy Policy .

Reviewer #2: No

---

## [Author Response · Author response to Decision Letter 2]

24 Jun 2025

Dear Editor,

Thank you very much for your minor review. I have highlighted the significant findings in Tables 2 through 4.

I hope the manuscript is suitable to accept for publication in PLOS One.

Thanks for being very positive about the submission.

Best regards

Amirul Islam

---

## [Editor Report · Decision Letter 2]

Association of lifestyle and behavioral factors with self-reported visual problems among schoolchildren in rural Bangladesh.

PONE-D-25-00919R2

Dear Dr. Islam,

We’re pleased to inform you that your manuscript has been judged scientifically suitable for publication and will be formally accepted for publication once it meets all outstanding technical requirements.

Kind regards,

Nick Fogt

Academic Editor

PLOS ONE

Additional Editor Comments (optional):

Thank you for your highlighting the significant findings.
---

## [Editor Report · Acceptance letter]

PONE-D-25-00919R2

PLOS ONE

Dear Dr. Islam,

I'm pleased to inform you that your manuscript has been deemed suitable for publication in PLOS ONE. Congratulations! Your manuscript is now being handed over to our production team.

Kind regards,

on behalf of

Dr. Nick Fogt

Academic Editor

PLOS ONE